# Entorhinal cortex receptive fields are modulated by spatial attention, even without movement

Niklas Wilming[1,2,3]*, Peter König[1,3], Seth König[2], Elizabeth A Buffalo[2]

[1]Institute of Cognitive Science, University of Osnabrück, Osnabrück, Germany; [2]Department of Physiology and Biophysics, University of Washington School of Medicine, Washington National Primate Research Center, Seattle, United States; [3]Department of Neurophysiology and Pathophysiology, University Medical Center Hamburg-Eppendorf, Hamburg, Germany

**Abstract** Grid cells in the entorhinal cortex allow for the precise decoding of position in space. Along with potentially playing an important role in navigation, grid cells have recently been hypothesized to make a general contribution to mental operations. A prerequisite for this hypothesis is that grid cell activity does not critically depend on physical movement. Here, we show that movement of covert attention, without any physical movement, also elicits spatial receptive fields with a triangular tiling of space. In monkeys trained to maintain central fixation while covertly attending to a stimulus moving in the periphery we identified a significant population (20/141, 14% neurons at a FDR <5%) of entorhinal cells with spatially structured receptive fields. This contrasts with recordings obtained in the hippocampus, where grid-like representations were not observed. Our results provide evidence that neurons in macaque entorhinal cortex do not rely on physical movement.

DOI: https://doi.org/10.7554/eLife.31745.001

*For correspondence:
nwilming@uke.de

Competing interests: The authors declare that no competing interests exist.

## Introduction

Spatial representations, in the form of place cells and grid cells, have been identified in rodents (*Hafting et al., 2005*; *O'Keefe and Dostrovsky, 1971*), bats (*Yartsev et al., 2011*; *Yartsev and Ulanovsky, 2013*), macaque monkeys (*Killian et al., 2012*) and humans (*Jacobs et al., 2013*; *Ekstrom et al., 2003*). The hippocampal formation, however, also contributes to the processing of memories (*Squire and Wixted, 2011*), and conceptual similarities between spatial navigation and the processes involved in remembering and planning suggest that grid cells might support cognitive functions besides spatial navigation (*Buzsáki et al., 2013*). This idea resonates well with the recent demonstration that grid cells are active during the exploration of images with eye-movements, that is by overt attention (*Killian et al., 2012*). These findings revealed that grid cells do not only track an animal's location in space, but can also represent the gaze position of the animal. Shifts of gaze location usually correspond to shifts in attention (*Hoffman, 1998*; *Rolfs et al., 2011*), and the neural substrate for overt and covert attention is largely overlapping (*Corbetta et al., 1998*). Accordingly, grid cells in the entorhinal cortex are potentially capable of representing not only gaze location but the locus of attention in general. Here, we investigated whether firing fields of entorhinal cells are activated by movements of covert attention, in the absence of any physical movement. Attention functions as an important control for mental processes (*Petersen and Posner, 2012*); accordingly, the hypothesis that grid cells participate in a range of cognitive functions predicts that grid cells may be activated by movements of attention. Here, we tested this prediction by recording from single units in the entorhinal cortex in monkeys trained to perform a task of covert attention.

**eLife digest** When driving home from work, we can easily tell where we are along our route. A network of regions deep within the brain acts a little like an inbuilt GPS and tracks our position relative to other objects in the environment. Cells in one of these brain regions, called the entorhinal cortex, construct a map of our surroundings. The map resembles a grid of tessellating triangles, and the cells that produce it are called grid cells. Whenever we move onto a corner of one of the triangles, the corresponding grid cells start to fire. This firing enables the brain to track our movements.

But grid cells may do more than support navigation through the physical world. They may also help us navigate through our memories. Evidence suggests that we organize our knowledge and experiences into mental maps. These maps connect related concepts just as physical maps connect nearby locations. In principle, the brain systems that support physical navigation may also enable us to access the information within our mental maps. But for this to be true, cells such as grid cells must respond to movement in mental space as well as in physical space.

Wilming et al. have now tested this possibility by training macaque monkeys to detect a color change in a dot moving across a screen. The monkeys had to keep their eyes fixed on the center of the screen throughout the task. This meant they could not move their eyes to track the dot. Instead they had to mentally move the focus of their attention. Wilming et al. monitored the activity of cells in the entorhinal cortex as the monkeys mentally followed the dot. As predicted, some of the cells fired in a triangular grid-like pattern similar to that seen when animals move through their environment. Grid cells can thus fire in the absence of physical movement.

These findings bring us closer to understanding how brain circuits that code for spatial locations also support other mental processes. Evidence from many sources suggests that structures within the brain's medial temporal lobe, including the entorhinal cortex, contribute to memory. The next challenge is to identify how these brain regions encode spatial and other types of information and use them to form memories.

DOI: https://doi.org/10.7554/eLife.31745.002

## Results

To examine spatial representations of cells in the entorhinal cortex during the movement of covert attention, we recorded the neuronal activity of 141 neurons in the entorhinal cortex of two rhesus macaque monkeys performing a covert attention tracking task. The monkeys were trained to maintain central fixation while attending to a small (1°) dot that moved around the computer monitor (see *Figure 1*). The task of the monkey was to respond by quickly releasing a response bar when the dot changed color. The color change occurred after a random time interval (700 – 2000 ms) following trial onset. Over the course of a recording session, the dot traced out several Hamiltonian cycles, thereby ensuring that in the limit of infinitely many cycles each screen location was visited equally often. Monkeys performed the task at >70% accuracy, with the majority of errors due to saccades to outside of the fixation window. Depending on their motivation, monkeys completed between two and eight cycles per session.

In order to examine spatial representations among entorhinal neurons, we evaluated the standard gridness measure (*Solstad et al., 2008*; *Langston et al., 2010*; *Brandon et al., 2011*) along with a novel analysis of the depth of firing rate modulation within the firing fields. Grid cells possess a regular firing field with increased firing rates at the nodes of equilateral triangles that evenly tile space. Importantly, tiling space with equilateral triangles leads to 2D autocorrelation functions with six peaks arranged in a hexagonal structure around the center peak. Whether or not a cell's firing field shows this regular pattern is usually evaluated by quantifying the 60° rotational symmetry of the 2D autocorrelation (*Solstad et al., 2008*; *Langston et al., 2010*; *Brandon et al., 2011*). An important property of the gridness score is that the autocorrelation normalizes away the absolute firing rate modulation over space. That is, a cell that shows only a weak change in firing rate as a function of space might have the same gridness as a cell that shows a strong change in firing rate. The gridness score is therefore susceptible to noise, and homogeneous firing fields might produce high gridness scores by chance. Furthermore, for downstream read out of the grid signal the size of firing rate

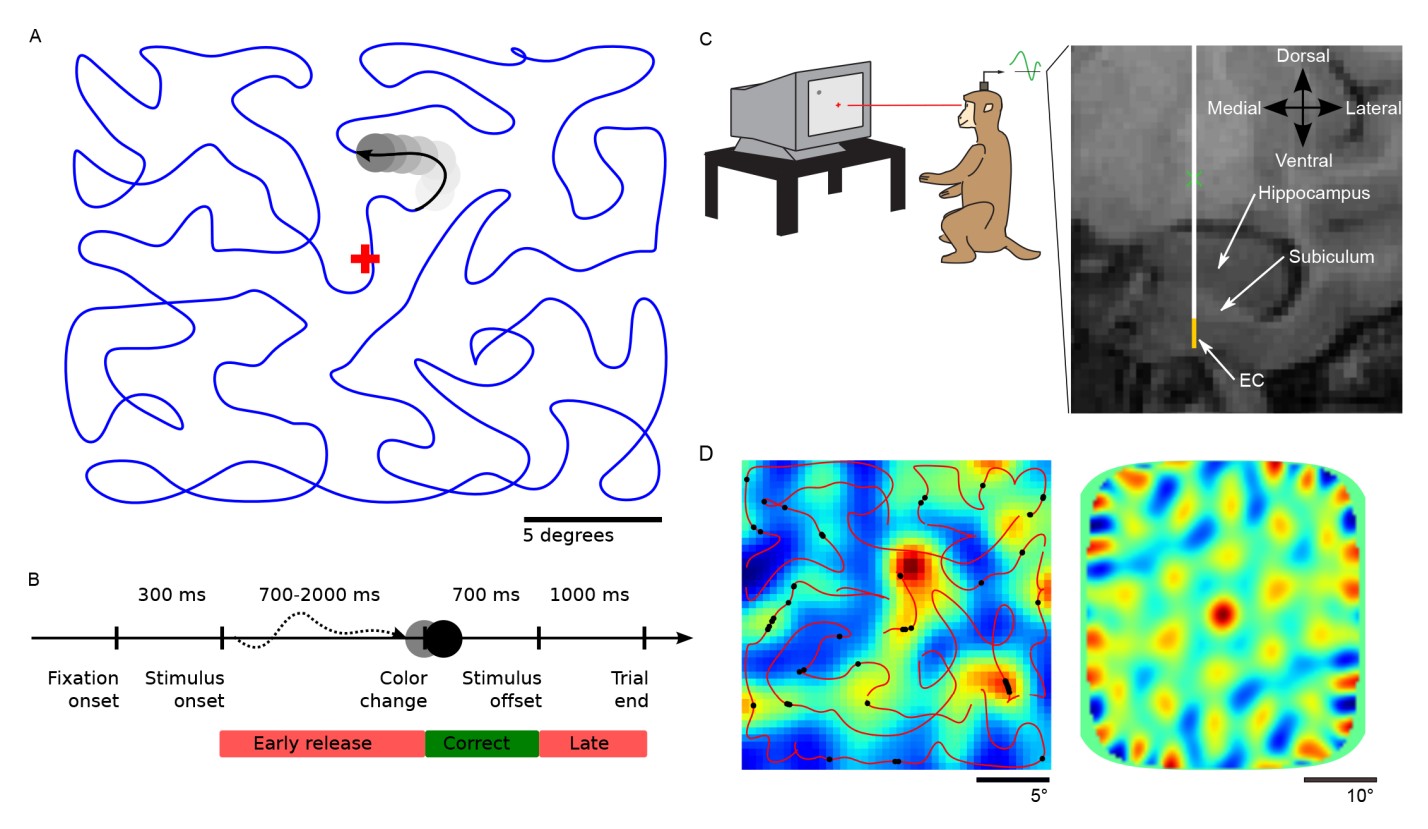

**Figure 1.** Task design and recording location. (**A**) Monkeys were trained to maintain central fixation while a dot moved across the screen and were rewarded for releasing a response bar when they detected a color change of the moving dot. Dot trajectories were smoothed Hamiltonian cycles through a 23 × 23 grid and covered the screen without a central bias. The blue trajectory was not shown during the experiment. (**B**) Temporal structure of a trial. (**C**) Recording setup and example of MRI-guided electrode placement (coronal section through the EC). Recordings were carried out with a 12-site laminar electrode array mounted on a tungsten microelectrode (AXIAL Array, 30 µm diameter, 150 µm spacing, FHC Inc.). Tungsten electrode is represented in white, the yellow strip represents the span of the recording contacts. EC = Entorhinal Cortex. (**D**) Example cell recorded under conditions of covert attention. Left panel shows the estimated firing field (max. firing rate 1.47 Hz). The small black dots on the trajectory indicate the spikes recorded during the first cycle. They largely align with the average firing field (backdrop) based on all six cycles recorded for this neuron. The 2D autocorrelation in the right panel shows six peaks surrounding the center which is characteristic of grid cells (gridness score 1.80).
DOI: https://doi.org/10.7554/eLife.31745.003

modulation, for example the signal to noise ratio is relevant. To better differentiate between noisy and grid-like firing fields, we computed for each cell an index of firing field modulation from its estimated rate map and a gridness score from its 2D autocorrelation function (see Materials and methods).

To assess the statistical significance of spatial representations in our recorded neurons, we computed the same indices on simulated cells with grid, place, or homogeneous firing fields. Cells were simulated by creating spike-trains with an inhomogeneous Poisson (IP) process (*Figure 2*). The rate function of the IP process was determined by the dot's trajectory through a firing field and a noise parameter that varied the influence of noise for each simulated cell (see Materials and methods). Simulated cells with a homogeneous firing field therefore control for any potential grid-like structure in the dot's trajectory. The resulting joint distributions of gridness scores and firing field modulation indices represent the likelihood of observing particular patterns of results. To differentiate grid cells, we computed the log likelihood ratio for comparing grid cells with place cells and homogeneous cells (*Figure 3* and Materials and methods). The resulting log likelihood ratio expresses how much more likely it is that a specific gridness score and firing field modulation index combination is produced by a grid cell compared to homogeneous or place cells. To control our false-discovery rate, we determined the appropriate threshold for classification performance on simulated non-grid cells. Out of 141 recorded neurons, we identified 20 cells (14%) with a log likelihood ratio of >4.25

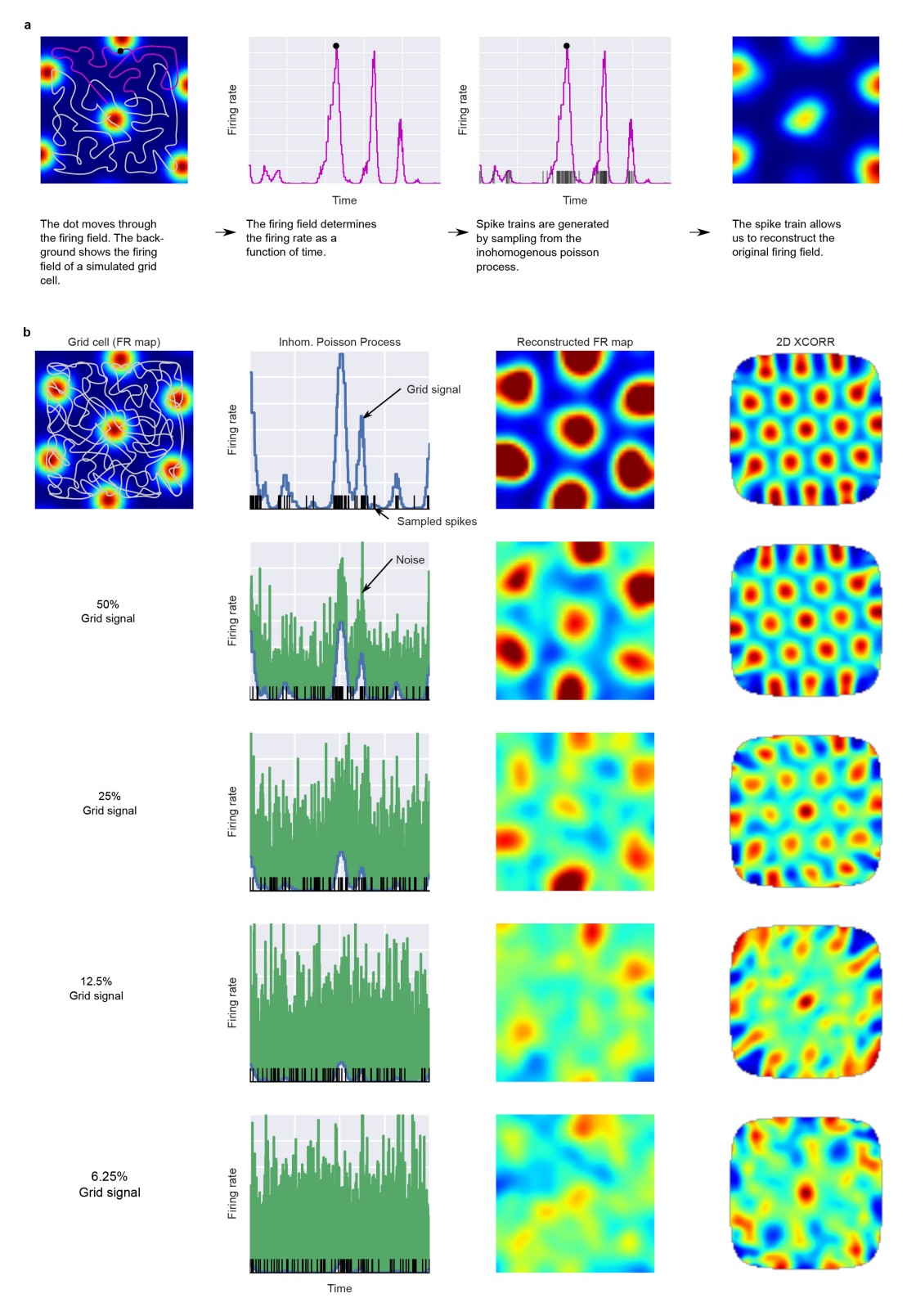

**Figure 2.** Simulation of units from grid-like firing fields. Panel (**A**) shows how the dot's trajectory and a 2D firing field generate the time varying firing rate for a simulated unit. Panel (**B**) shows how different amounts of noise influence the reconstructed firing fields and 2D autocorrelations.

DOI: https://doi.org/10.7554/eLife.31745.004

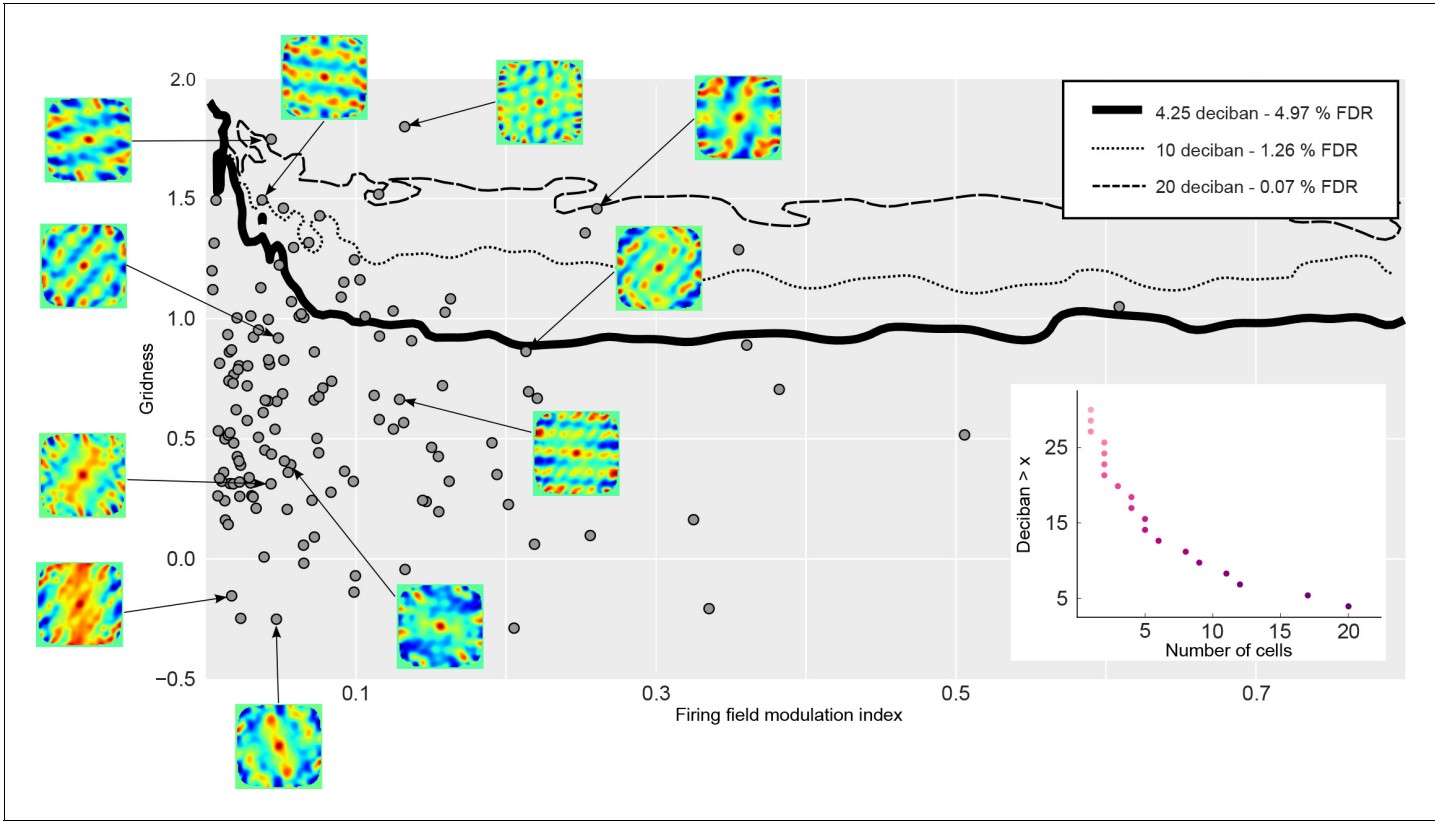

**Figure 3.** Classification of grid cells. Gray filled circles indicate gridness and firing field modulation index values for our recorded EC units. Solid curves show contours of the log likelihood ratio (in deciban) that indicates evidence in favor or against a cell being a grid cell. 4.25 deciban (thick line) corresponds to a false discovery rate (FDR) of <5% in our simulations, 10 deciban (10:1 odds ratio, thin dotted line) to a FDR of 1.26% and 20 deciban (thin dashed line) to a FDR of 0.07%. The FDR indicates how many simulated non-grid cells are classified as grid cells with each log likelihood ratio value. Cells above the curves are likely grid cells. Small insets with arrows show the 2D autocorrelation for example cells. Inset panel shows how many cells exceed a specific deciban threshold.

DOI: https://doi.org/10.7554/eLife.31745.005

deciban (corresponding to a false discovery rate <5%) in favor of cells with spatially structured firing fields according to the two criteria of modulation index and gridness score. Cells classified as grid cells had a lower mean firing rate (0.47 Hz) compared to those classified as other cells (1.22 Hz), and higher average gridness (1.29 vs. 0.51) and firing field modulation index values (0.1 vs. 0.04). We also tested more conservative thresholds of 10 and 20 deciban (false discovery rates of 1.26% and 0.07%, respectively), which require more evidence to classify a cell as a grid cell. Regardless of the threshold used, we were able to identify a population of neurons that were labeled as grid cells on an individual basis with high significance ($X^2(1, N = 141, FDR = 0.05) = 25.0$, $p < 0.0001$, 20, 9 and 3 cells w.r.t. to false discovery rates for different thresholds). We also identified candidate cells that qualitatively resembled grid cells but which could also be explained by assuming a noisy place like firing field (lower right points in *Figure 3* ).

We, furthermore, sought to validate our novel statistical approach with two different control analyses. First we recorded neuronal activity of 307 neurons from the hippocampus in monkeys performing the same task. There are currently no reports of grid cells in the hippocampus and these neurons can therefore serve as a baseline to determine our empirical false discovery rate. Classifying these cells with the same measures and the 5% FDR threshold yielded an empirical false-discovery rate of 8.5%. The FDR is slightly higher than the FDR derived from simulations because the hippocampal recordings likely contain a different ratio of spatially structured (e.g. place cells) to spatial noise cells. Yet, in the entorhinal cortex (see above), we find significantly more grid-cells than expected from a FDR of 8.5% ($X^2(1, N = 141, FDR = 0.085) = 5.9$, $p = 0.016$; $p < 0.05$ for classifying with the 10 and 20 deciban thresholds as well). Second, we used a bootstrapping analysis to create a joint

distribution of indices when the association between firing rate and the dot's position is destroyed. To this end, we estimated the firing rate of each neuron along the dot's trajectory and shuffled trajectory segments of 50 ms length (700 permutations). We again computed both indices and found an empirical false discovery rate of 5.5%. These control analyses suggest that our main analysis provides an adequate control for the false discovery rate.

We calculated two further controls to eliminate the possibility that our results could be due to the specific set of trajectories used in each session. First, combining spikes generated by different firing fields destroys the spatial structure of the individual firing fields. Any residual grid-like spatial structure of a pooled unit is therefore due to the specific stimulus conditions. We therefore pooled all neurons in each recording session and calculated gridness scores and firing field modulation indices. None of these pooled units was classified as having a grid-like firing field (average and max gridness score 0.36/0.96, average and max firing field modulation 0.02/0.11). Second, we investigated the spatial periodicity of trajectories in isolation. For each recording session, we pooled all trajectories and computed gridness scores and firing field modulation indices by assuming an equal firing rate at all locations. None of these units was classified as a grid cell (average and max gridness score 0.1/ 1.17, average and max firing field modulation 0.018/0.033). These controls demonstrate that our results are not due to an inherent grid-like structure in the trajectories.

As a last control, we examined whether our results might be confounded by the monkeys overtly tracking the dot with small eye-movements inside the fixation window within a trial. To investigate this, we compared the distance of the eye to the fixation cross with the dot's distance to the fixation cross. If eye-movements tracked the dot's position, these signals should be strongly correlated in each trial. However, we found no substantial correlations between eye-position and dot position. Average correlations within a session were small (mean = −0.005) and within the expected range of correlations when we shuffled trials in time and thereby assigning eye-movement positions to dot trajectories from different parts within a session (mean 0.001, 5 and 95% percentile: −0.02, 0.02). We conclude that the monkeys did not track the dot with small eye movements.

Systematic biases in receptive field size and position caused by hemisphere or eccentricity specific processing should show up as systematic biases in firing rate across the visual field. We therefore computed two asymmetry indices. The first quantifies the difference in firing rate between the right and left side of the visual field. While cells classified as grid cells (5% FDR) in both animals showed, on average, increased firing rates on the contralateral side of the visual field relative to the recorded hemisphere, these differences were not consistent across cells (mean contralateral-ipsilateral = 0.06, t = −1.99, p = 0.06, one sample t-test). Second, we computed an index that compares average firing rate within the central five degrees of visual field around the fixation cross with the rest of the visual field. We again found no difference in firing rate (mean center-periphery = 0.08, t = 1.7, p = 0.10). These numbers, however, have to be taken with a grain of salt. In each monkey, we recorded from only one hemisphere, making an intra-monkey comparison of left/right visual field responses difficult. Furthermore, the experiment was not designed to optimize investigation of systematic changes with eccentricity. For such an experiment, monkeys trained to covertly track targets at high eccentricity and as well as having more recording time for each neuron with attention directed to eccentric locations would be desirable. These two constraints interact and make respective experiments highly challenging.

An important caveat is that our experimental conditions are most sensitive for firing fields with a size that allows multiple repetitions of the grid on the screen. Larger firing fields might appear like place or homogeneous cells, and firing fields with spacing smaller than the spacing of our trajectories will appear as spatially structured noise. We can therefore make no definitive statements about the percentage of grid cells in the entorhinal cortex, and our results likely represent a conservative estimate. Future studies could address this issue by a) using a larger screen to allow for larger field sizes and b) using fixation locations that are not centered to compensate for the loss of visual acuity at large eccentricities. For example, repeating the current experiment with a larger screen and additional fixation locations at the current screen corners would increase the covered area four-fold.

## Discussion

In summary, our results provide strong evidence that the activation of cells in entorhinal cortex with spatially structured firing fields does not require physical movement through an environment.

Previous results demonstrated that, in stationary monkeys, grid cells can be activated by eye movements as monkeys visually explore complex scenes (*Killian et al., 2012*). Here, our findings suggest that even eye movements are not required to activate cells with grid-like firing fields and that firing fields in the entorhinal cortex cells can represent the location of attention. That spatial representations in the hippocampal formation can represent locations other than the animal's current position has been shown in rodents through the identification of theta phase precession (*O'Keefe and Recce, 1993*) and the demonstration that ensembles of hippocampal place cells fire in sequence within a theta cycle, representing places in front of and behind the animal (*Muller and Kubie, 1989*; *Skaggs and McNaughton, 1996*; *Jensen and Lisman, 1996*). More recent studies have shown that sequential firing of hippocampal place cells can represent anticipation of upcoming locations (*Diba and Buzsáki, 2007*), future possible paths when the rat is at a choice point (*Johnson and Redish, 2007*; *Singer et al., 2013*), and even behavioral trajectories toward a remembered goal in a 2D environment (*Pfeiffer and Foster, 2013*). While it is difficult in rodents to assess the location of attention distinct from the location of the animal, these data are consistent with the idea that hippocampal sequences may reflect the animal's attention being directed along a specific spatial trajectory. Paying attention to the just-completed trajectory could serve to enhance encoding of that experience. Correspondingly, directing attention to future possible paths could serve as a retrieval mechanism to enable optimal choice behavior. These findings also resonate well with reports of grid-like six-fold symmetric BOLD activity while participants imagine navigating in a familiar environment (*Horner et al., 2016*), imagine directions between locations in a large-scale virtual city (*Bellmund et al., 2016*), and navigate a purely abstract concept space (*Constantinescu et al., 2016*). The data presented here demonstrate that entorhinal cells with grid-like firing fields can similarly encode information apart from the animal's current location.

In agreement with *Killian et al. (2012)*, we found a percentage of grid cells (14%) that is lower than reports from recording in rats (e.g. 32% in *Sargolini et al., 2006*). We see several possible explanations for this discrepancy. First, our task design is limited to detecting a subset of grid cells who's firing fields match the size of the screen and the spacing of the dot's trajectory, thereby potentially underestimating the true percentage of grid cells in macaque entorhinal cortex. Second, our task design does not attach significance to the spatial position of the dot. It is possible to solve the task without knowing where the color change occurs. Thus, it seems plausible that our observations reflect the entorhinal cortex in an attenuated state and that we would observe stronger grid cell activity if the precise dot position was relevant. Future work could address these points by enlarging the screen, including more fixation locations around which the moving dot needs to be observed, and by making the spatial position relevant. The latter might be achieved by introducing an invisible spatial reward field, such that the monkey does not need to detect a color change but only respond when the dot passes through high reward areas on the screen. A good strategy would thereby require integration of the spatial position of received reward to infer the layout of the reward field. Future experiments examining links between grid cell responses and behavior are critical for advancing our understanding of the function of the entorhinal cortex.

## Materials and methods

### Summary

Each recording session consisted of the presentation of several cycles of the moving dot. Cycles were smoothed Hamiltonian cycles through a 23 × 23 grid with 1°spacing and therefore showed no center bias. Each cycle consisted of an average of 114 trials and took the monkeys an average of 13 min to complete. The speed of the dot changed as a function of the trajectory's curvature, slowing down during corners and speeding up during straight parts. The average speed of a typical trajectory was 1.95°/s, with a maximum of 3.75°/s. Each trial ended with the correct detection of the color change, a missed detection, or when the monkey looked outside of the 4.5° × 4.5° central fixation window. In subsequent trials, the dot reappeared at its ending location from the previous trial. After the completion of a cycle, the next cycle was chosen randomly from a set of 100 precomputed trajectories. Excluding fixation errors monkeys performed the task at 85% accuracy across all sessions (min = 70%, max = 94%, std = 8%) and 87% accuracy across cycles (std = 13%). We only analyzed data from trials with correct detection of the color change and constant fixation within the fixation

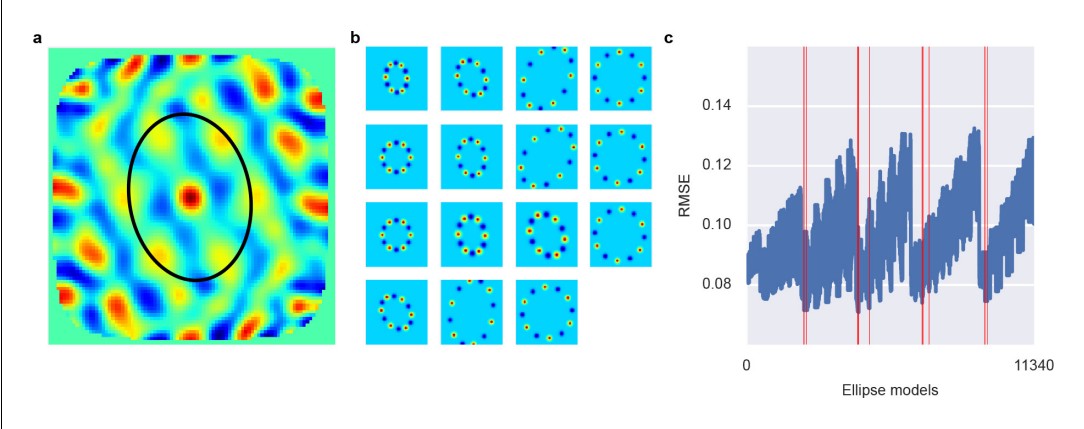

**Figure 4.** Ellipse fitting on 2D autocorrelation maps. (A) shows one example 2D autocorrelation function and fitted ellipse that passes through six peaks. (B) shows a set of candidate ellipses with hexagonal peaks and valleys. To identify best-fitting ellipses, we generated a large number of potential ellipses that passed through at least one peak in the 2D autocorrelation and the computed the RMSE between ellipses and 2D autocorrelation. (C) shows the RMSE of a large number of candidate ellipses, arranged by a linear index into the five-dimensional parameter space for the ellipses. Red lines indicate ellipses selected by a peak shift algorithm whose maximum gridness score was assigned to this unit.

DOI: https://doi.org/10.7554/eLife.31745.006

window. The monkeys were head-fixed and seated in a chair such that the monitor's center corresponded to their neutral eye position. Spikes were recorded (40 kHz) from the entorhinal cortex with a laminar electrode array mounted on a tungsten microelectrode (12-site, 150-µmm spacing; FHC). Rate maps were computed with a standard Gaussian smoothing procedure (*Killian et al., 2012*; *Brandon et al., 2011*).

Gridness scores were calculated using standard equations (5,14–16), but we additionally validated that our implementation was highly accurate in distinguishing grid from non-grid cells in simulated data (see 'Gridness score implementation', 'Unit simulation', 'Log likelihood ratios and classification of units' and methods *Figures 2,4,5*). We computed a firing field modulation index to separate spatially structured firing fields from homogeneous noise-dominated fields. The firing field modulation index captures how much of the spatial variation in firing rate can be explained by the mean firing rate over space:

$$ffm(g) = \left( \sum_i g_i^2 - \sum_i \mu(g)^2 \right) / \sum_i g_i^2$$

where g is the two-dimensional rate map which is linearly indexed by i and µ (g) is the mean firing rate. Significance was established by computing the $\log_{10}$ likelihood ratio between the likelihood that a specific cell's gridscore and firing field modulation index was produced by a noisy grid-cell or a cell with a noisy homogeneous or noisy place field like firing field (see 'Log likelihood ratios and classification of units'). All experiments were carried out in accordance with protocols approved by the Emory University and University of Washington Institutional Animal Care and Use Committees.

## Behavioral task

We recorded neuronal activity in the entorhinal cortex in one male and one female monkey (maccaca mulatta) trained to perform a covert attention tracking task. Before each session, the monkey's chair was positioned such that the eye-screen distance was 60 cm. Monkeys were head-fixed during recordings such that this distance never changed. Reward was delivered through a tube positioned in front of the monkey's mouth and reward delivery was always accompanied by an auditory beep. The task required the monkeys to attend to a dot as it moved across the screen. The monkey's task was to maintain central fixation within a 4.5 × 4.5° window and to release a bar when the moving dot changed color. Gaze location was monitored with an infrared eye-tracking system (ISCAN, Inc.), whose camera was placed underneath the screen. Each trial began with the onset of a central fixation cross. After the monkey fixated the cross for 300 ms, a moving dot appeared. The dot moved

for a random duration between 700 and 2000 ms before it changed color (from light gray to white on a dark grey background). After the color change, a bar release that occurred within 700 ms was counted as a successful detection. Early and late bar releases as well as fixational breaks were counted as errors. Upon bar release or 700 ms after the color change the moving dot disappeared. The monkey was rewarded with a food-based slurry on successful trials. In the next trial, which started 1 s after reward delivery, the dot continued on the same trajectory, i.e. starting from where it had disappeared in the previous trial. Monkeys worked for as long as they were well-motivated and completed between two and eight cycles per session.

Monkeys were taught the covert attention task with the following sequence of simpler tasks. First, monkeys were trained in a fixation task in which they were rewarded for keeping fixation for some duration on the central fixation cross. As soon as behavior in the fixation task was stable, the dot was introduced behind the fixation cross and monkeys were now rewarded for releasing a bar when the dot changed color. As part of their general training and habituation monkeys had previously been taught a similar color change task, allowing them to transition to this task. In the beginning, the dot changed color after short durations (300–500 ms), which were prolonged until the final duration was achieved. Next, the experimenter manually moved the dot in small steps away from the fixation cross. This was practiced until monkeys would reliably fixate the fixation cross and detect color changes of the.

The dot followed trajectories that visited different parts of the screen only once during the course of an entire cycle. Such trajectories are called Hamiltonian cycles and ensure that the distribution of dot locations is not biased to specific locations. To create such trajectories, we covered the screen with a matrix of 23 × 23 nodes (1° spacing) that evenly covered an area of −11:11° of visual angle on the screen. We conceptualized this matrix as a graph where neighboring grid nodes are connected such that the dot can move along the vertices of this graph. Creating a Hamiltonian cycle now amounts to finding a sequence that visits all nodes in this graph exactly once. This problem is identical to the traveling salesman problem. Accordingly, each trajectory was created by using the Concorde (http://www.math.uwaterloo.ca/tsp/concorde/) traveling salesman problem solver to find a path that visits each node of the 23 × 23 graph exactly once. We changed the random seed for each sequence to generate unique sequences. Next, we linearly connected all graph nodes to create

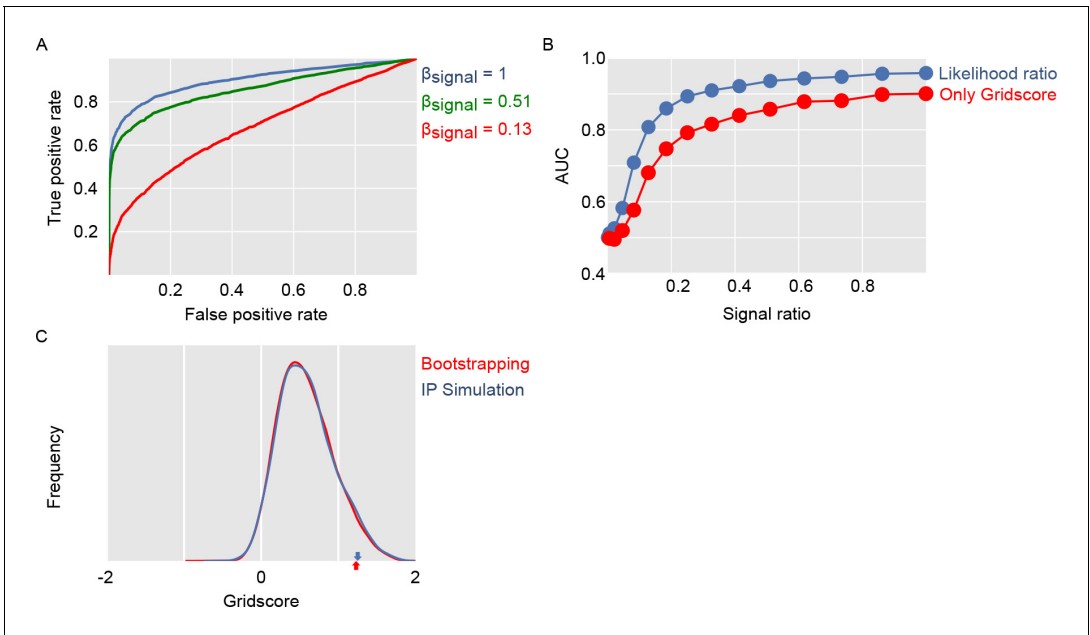

**Figure 5.** Evaluation of gridness score implementation and log likelihood ratio classification. Panel (**A**) shows ROC curves for separating simulated grid-like units and simulated units with homogeneous firing fields. Panel (**B**) compares using only the gridness score with our approach of computing log likelihood ratios based on the ffm and gridness score. Panel (**C**) compares gridness scores obtained by shuffling spike sequences along the dots trajectory with gridness scores obtained from simulating units with homogenous noise.
DOI: https://doi.org/10.7554/eLife.31745.007

path through the graph. The x and y coordinates of this trajectory were smoothed with a Gaussian kernel with σ = 60° to yield a smooth trajectory. The smoothing also ensured that the dot continuously changed its speed (mean for a typical trajectory 1.95°/s, 3.75°/s maximum). At the beginning of a recording session and after completion of each cycle, the next trajectory was chosen randomly from a precomputed set of 100 trajectories. Stimuli were displayed on a 19' CRT monitor with a refresh rate of 120 Hz.

## Electrophysiological recordings

Spikes (40 Khz) were recorded from the EC with a laminar electrode array mounted on a tungsten micro electrode (AXIAL array: 12-site, 30 μm diameter, 150 μm spacing, FHC, Inc.) with recording hardware from Plexon Inc. Subsequent analysis was carried out with custom MATLAB (The MathWorks, Natick, MA) and python code (https://github.com/nwilming/ovtcvt [*Wilming, 2018*]; copy archived at https://github.com/elifesciences-publications/ovtcvt). Recordings from monkey MP (male) were carried out in the right hemisphere and recordings from monkey PW (female) were in the left hemisphere. Recording sites were planned with the help of MRI scans of each monkey and an atlas. Recording sites were located in the posterior part of the entorhinal cortex and close to the rhinal sulcus. Each recording started by slowly lowering a stainless steel guide tube containing the laminar array through a craniotomy. The array was then slowly advanced to its final position while the monkey conducted unrelated tasks. Recordings were started a few minutes (~10) after the array reached its final position to allow the tissue around the electrode to settle. Spikes were sorted using offline methods (OfflineSorter, Plexon Inc.). We discarded units that did not show stable firing behavior through an entire cycle. In total, we analyzed 69 units from monkey MP (seven sessions) and 72 units from monkey PW (eight sessions). See *Table 1*.

Neural recordings from the hippocampus were conducted in a similar manner. Four independently moveable tungsten microelectrodes (1 to 3 MΩ, FHC Inc.) were lowered down into the hippocampus with the use of coordinates derived from MRI scans. Recordings took place in the left anterior portion of the hippocampus in one monkey (PW) and in the right posterior portion of the hippocampus in another monkey (TO). Neurons were recorded from CA1-CA4 subfields, dentate gyrus, and subiculum. For the hippocampal recordings, we recorded from the covert attention task for two to six cycles per session following a ~ 1 hr recording during which monkeys freely viewed images.

## Rate maps and autocorrelation

Rate maps and autocorrelation functions were computed analogously to *Killian et al. (2012)*. We computed 2D histograms of spike and dot positions with bins of size 0.5° × 0.5°. To estimate a firing rate map, we smoothed both histograms with a Gaussian kernel (σ = 1.25°) and divided the spike counts by the amount of time the dot spent in the respective locations. 2D autocorrelations were computed by shifting the firing rate map relative to itself and computing the coefficient of correlation for the overlapping maps.

## Statistical analysis

Our statistical analysis proceeded in several steps. Our goal was to classify the reconstructed firing fields of recorded units as grid-like or not grid-like. In a first step, we identified two measures of firing behavior that were suitable for this classification. In a second step, we estimated the joint distribution of these measures when the generating cell is either a noisy grid cell, a cell with a noisy homogeneous firing field or a noisy place field like cell. This allowed us to compute a log likelihood ratio that indicates whether or not a firing field is caused by a grid cell or by the other types of simulated cells. We then used the log likelihood ratio for classification of cells as grid cells or other cells.

**Table 1.** Number of recorded units (included/excluded) for each monkey and area.

|  | MP | PW | TO |
|---|---|---|---|
| *EC* | 69/16 | 72/25 | —/— |
| *Hippocampus* | —/— | 97/3 | 189/18 |

DOI: https://doi.org/10.7554/eLife.31745.008

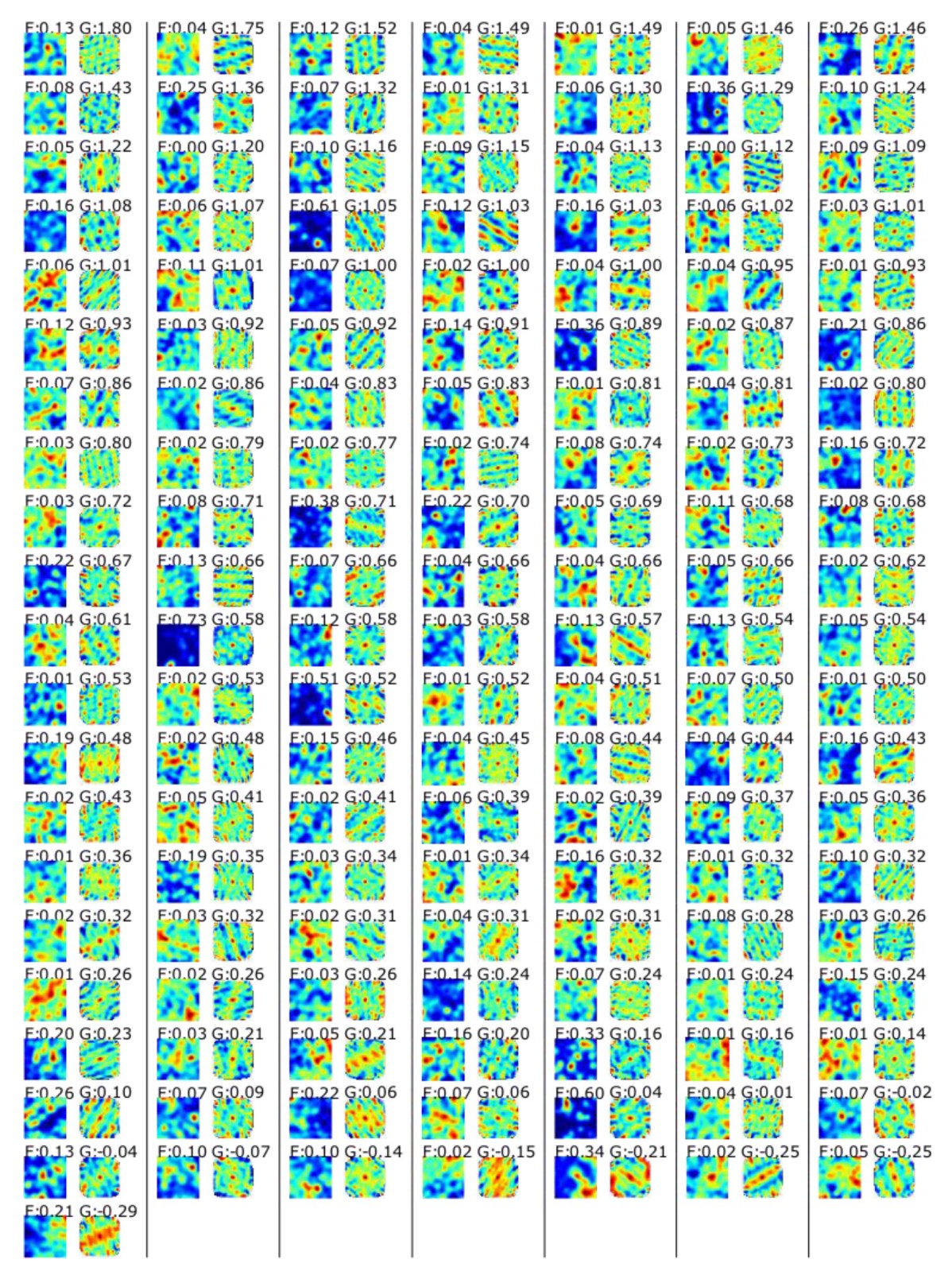

**Figure 6.** Rate maps (left) and autocorrelation functions (right) for all recorded EC cells. Numbers on top give the firing field modulation index (F) and the gridness score (G).

DOI: https://doi.org/10.7554/eLife.31745.009

Both of these steps made extensive use of spike trains of simulated neurons. We therefore first describe our model for simulating spike trains and then describe the classification process.

## Unit simulation

We simulated units by assuming that firing behavior is the result of an inhomogeneous Poisson process (IP). This process allows to model different firing rates for different positions of the moving dot. The rate function of the IP is determined by a firing field, through which the dot passes, and a noise component. Sampling from the IP defined by this rate function yields artificial spike trains caused by the underlying noisy grid field. We simulated spike trains from noisy grid-like, noisy homogeneous or noisy place field-like firing fields. Grid-like firing fields were created by placing

$$\lambda(x,y) = \sum_{p \in P} \mathcal{N}\left((x,y), gt\left[(p_x + o)\cos\theta - p_y \sin\theta, gt(p_x + o)\sin\theta + p_y \cos\theta\right], gt\sigma\right)$$

$$P = \left\{\sqrt{3}iDS, jD\right\} \cup \left\{\sqrt{3}(iDS + S\frac{D}{2}), jD + \frac{D}{2}\right\} \forall i \in \mathbb{Z}$$

2D Gaussian distributions at the nodes of a grid that tiles space with equilateral triangles: where $\lambda(x,y)$ is the firing rate of the grid field at location $(x,y)$, $D$ is the distance between peaks, $S$ is a scaling parameter, $\theta$ rotates the grid field, $o$ shifts the grid field and $\sigma$ specifies the size of each grid node. The grid component of the rate function represents the firing rate in the firing field along several randomly drawn dot trajectories. The dot trajectories transform the 2D firing field into a 1D time- and space-varying firing rate. Simulated units were evaluated with the same pool of trajectories used for the recorded units. The noise component of the rate function is drawn from an exponential distribution (rate = 1) for each location in the trajectory. The amount of noise and grid signal in the time

$$\lambda(t) = R\left[\beta_{signal}\lambda_{Grid}(t) + (1 - \beta_{signal})\lambda_{Noise}(t)\right]$$

and space varying firing rate is governed by the parameter $\beta$ signal : $\lambda_{Grid}$, $\lambda_{Noise}$ are the grid- and noise-dependent components of the firing rate and are scaled to have an expected firing rate of 1. $\beta_{signal}$, with $0 < \beta_{signal} < 1$, specifies how much the grid signal contributes to the firing rate. R is the desired mean firing rate (here uniformly sampled from the empirical firing rates). $\lambda(t)$ is therefore the firing rate of the IP. Spike trains were then generated by producing a sample from the IP defined by the time- and space-varying firing rate $\lambda(t)$. The rate function has a temporal resolution of 120 Hz. Methods *Figure 2* shows a graphical overview of unit simulations.

We simulated 5000 grid cell units for each of nine different $\beta_{signal}$ values ($i^2$ for evenly spaced i between 0.18 and 1). All other parameters were drawn from random distributions:

$$
\begin{aligned}
D &\sim & u(7,16) \\
\sigma &\sim & u(1,2) \\
\theta &\sim & u(0,90) \\
S &\sim & min(1.5, max(0.5, N(1, \tfrac{1}{8}))) \\
C &\sim & u(2,6) \\
o &\sim & u(-2.5, 2.5)
\end{aligned}
$$

where C is the number of dot trajectories used for the simulation. Parameters are given in degrees of visual angle and ranges were chosen to allow broad coverage of potential grid-like firing fields. We discarded all firing fields that had less than three peaks in the area covered by the trajectory; these are a priori not recognizable as grid-like. This resulted in 27,450 units whose firing rate was determined by a grid-like firing field. We also simulated units based on spatially structured firing fields that were non grid-like. We used place-like firing fields that had only one peak by setting the spacing parameter of the grid-like firing fields to larger than twice the screen size. The standard deviation was randomly drawn from the interval [3,4] to allow large parts of the trajectory to be influenced by the place field. The scaling was set to 1. We used 15 $\beta_{signal}$ values ($i^2$ evenly spaced i between 0 and 1) and simulated 2500 units with each $\beta_{signal}$ value. This resulted in 37,500 simulated units whose firing fields ranged from completely uniform ($\beta_{signal} = 0$) to spatially structured fields ($\beta_{signal} > 0$). To summarize, we simulated a large range of plausible grid fields that can be resolved

by our trajectories. In addition, we simulated cells with noisy homogeneous firing fields and cells with a noisy place-like firing field.

## Gridness score implementation

Computation of the gridness score is analogous to (*Langston et al., 2010*; *Brandon et al., 2011*). In this computation, an ellipse is fit around six peaks in the 2D autocorrelation function and the rotational correlation is computed at 30, 60, 90, 120 and 150 degrees. The gridness is then defined as:

$$\text{Gridness} = \min(\text{rc}(60), \text{rc}(120)) - \max(\text{rc}(30), \text{rc}(90), \text{rc}(150))$$

where rc($x$) refers to the rotational correlation at angle $x$. *Brandon et al. (2011)* noted that this measure is sensitive to how the ellipse is fit in the 2D autocorrelation. Choosing the closest six peaks, for example, might include a peak that is due to noise in the firing field and omits a 'correct' peak that is further away. We used a two-step procedure to fit an ellipse. In a first step, potential ellipses that likely contain six peaks and six valleys were identified. This was achieved by predicting for a large space of possible grid fields where the most central peaks and valleys would be in the 2D autocorrelation for this grid field. Grid fields were parameterized by grid spacing, aspect ratio, size of the peaks, rotation of the grid field and shift of the peaks along the ellipse. The

$$r(x,y) = \sum_{\alpha \epsilon P} N((x,y), \mu(\alpha + L), \sigma) + \frac{1}{2} \sum_{\alpha \epsilon P} N((x,y), \mu(\alpha + L + 30), \sigma)$$

$$P = \{0, 60, 120, 180, 240, 300\}$$

$$\mu(\alpha) = (t(\alpha)_x cos\theta - t(\alpha)_y sin\theta, t(\alpha)_x sin\theta + t(\alpha)_y cos\theta)$$

$$t(\alpha) = (-SDsin\alpha, Dcos\alpha)$$

prediction r(x,y) was computed according to the following equations: where D is the grid spacing, $\theta$ is the rotation of the grid field, $\sigma$ describes the size of the grid peaks, S relates to the aspect ratio of the grid field and L describes the shift of the peaks along the ellipse. We constrained the set of candidate ellipses to those that passed through at least one non-central peak in the 2D autocorrelation. Methods *Figure 4* shows several example predictions. The second step consisted of selecting ellipses that capture structure in the 2D autocorrelation. To this end, we computed the RMSE (root-mean-square error) between all ellipse predictions and the 2D autocorrelation functions. We then used a minimum shift algorithm to select ellipses that captured six peaks and valleys in the autocorrelation function better than ellipses in the local neighborhood of possible ellipses (the window of the minimum shift algorithm is N/5). The final gridness score was then the maximum of the gridness scores computed for these ellipses.

To evaluate the power of the gridscore implementation to separate grid cells from non-grid cells, we computed gridness scores for all simulated units with grid-like firing fields. We then compared how well the gridscore separated units with firing rates determined by noise ($\beta_{signal}$ = 0) from those whose firing rate was at least partly determined by the grid firing field ($\beta_{signal}$ >0). To this end, we computed ROC curves and used the area under the curve as a measure of discriminatory power. Methods *Figure 5* shows these ROC curves and the AUC as a function of $\beta_{signal}$.

## Log likelihood ratios and classification of units

To classify cells as grid-like we transform gridness scores and firing field modulation (ffm) index combinations into a single likelihood value. This mapping is derived via simulating many different random firing fields, place like firing fields or grid-like firing fields and computing gridness scores and ffm index values. This allows us to differentiate cells based on an estimate of the likelihood that a given gridness score and ffm index is generated by a grid-like firing field or other firing fields. To differentiate grid cells from non-grid cells, we computed the log likelihood ratio that compares the evidence that a gridscore and firing field modulation (ffm) combination is generated by a grid cell to the evidence that it is generated by a cell with a homogeneous or place field-like firing field:

$$\log_{10}(P_{grid}(gridscore, ffm)/P_{nongrid}(gridscore, ffm))$$

The probabilities $P_{grid}$ and $P_{nongrid}$ were computed from the simulation results with grid-like, place field-like and homogeneous firing fields. Since the gridness score and ffm index are continuous values, we locally pool simulated gridness scores and ffm indices to compute the likelihood. This

pooling is carried out by a Kernel density estimate and the bandwidth (size of pooling) is determined via 10-fold cross validation. We thus estimated each probability distribution with a kernel density estimator and used these estimates to derive the likelihood estimates. Since the gridness score and firing field modulation measures are bounded, we applied a logit transformation to avoid boundary artifacts. We only considered units with $\beta_{signal} > 0.15$ as simulations of grid cells for the grid density estimate. Smaller $\beta_{signal}$ values produced firing fields qualitatively close to those obtained with $\beta_{signal} = 0$ and thereby inflated the likelihood of observing small gridness scores and firing field modulation values from the grid cell simulation. Removing these units for the density estimation therefore makes our estimates more conservative. The log likelihood ratio expresses how much evidence we have that a particular gridness score and firing field modulation combination is produced by a grid cell relative to the evidence for place cells or cells with a homogeneous firing field.

To evaluate the power of the log likelihood approach to separate grid cells from non-grid cells, we compared how well the log likelihood value separated units with firing rates determined by noise ($\beta_{signal} = 0$) from those whose firing rate was at least partly determined by the grid firing field ($\beta_{signal}>0$). In analogy to our evaluation of different gridness scores, we used the area under ROC the curve as a measure of discriminatory power. Methods *Figure 5B* shows AUC as a function of $\beta_{signal}$ in comparison to only using the gridness score. The log likelihood approach performed better than only using the gridness score for all noise levels. *Figure 6* shows 2D autocorrelation functions and ratemaps for all EC cells.

## Acknowledgements

This study was supported by National Institutes of Health Grants MH080007 (EAB) and MH093807 (EAB), and the Office of Research Infrastructure Programs (ORIP) Grants P51OD010425 (Washington National Primate Research Center) and P51OD011132 (Yerkes National Primate Research Center), the FP7 project eSMCs IST-270212 (NW, PK) and SFB 936 (B6) Multi-Site Communication in the Brain (PK).

# Additional information

### Funding

| Funder | Author |
| --- | --- |
| National Institutes of Health | Elizabeth A Buffalo |
| Office of Research Infrastructure Programs | Elizabeth A Buffalo |
| European Commission | Niklas Wilming Peter König |
| Deutsche Forschungsgemeinschaft | Peter König |

The funders had no role in study design, data collection and interpretation, or the decision to submit the work for publication.

### Author contributions

Niklas Wilming, Conceptualization, Software, Formal analysis, Validation, Investigation, Visualization, Methodology, Writing—original draft, Writing—review and editing; Peter König, Conceptualization, Formal analysis, Supervision, Funding acquisition, Writing—original draft, Writing—review and editing; Seth König, Investigation, Writing—review and editing; Elizabeth A Buffalo, Conceptualization, Supervision, Funding acquisition, Writing—original draft, Project administration, Writing—review and editing

### Author ORCIDs

Niklas Wilming https://orcid.org/0000-0003-0663-9828
Peter König https://orcid.org/0000-0003-3654-5267

Seth König [iD] http://orcid.org/0000-0002-1600-0342
Elizabeth A Buffalo [iD] https://orcid.org/0000-0001-6326-9187

### Ethics

Animal experimentation: This study was performed in strict accordance with the recommendations in the Guide for the Care and Use of Laboratory Animals of the National Institutes of Health. All of the animals were handled according to approved institutional animal care and use committee (IACUC) protocols (#4316-01) of the University of Washington. The protocol was approved by the Office of Animal Welfare of the University of Washington (D16-00292). The Washington National Primate Research Center (WaNPRC) is accredited by the AAALAC (#000523) and registered with the USDA (91-R-0001). The University of Washington (UW) is committed to conducting quality animal research in an ethical and responsible manner to further science and to improve the health of society.

### Decision letter and Author response

Decision letter https://doi.org/10.7554/eLife.31745.012
Author response https://doi.org/10.7554/eLife.31745.013

## Additional files

### Supplementary files

• Transparent reporting form
DOI: https://doi.org/10.7554/eLife.31745.010

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
