## [Decision Letter]

Thank you for submitting your article "Entorhinal cortex receptive fields are modulated by spatial attention, even without movement" for consideration by *eLife*.

Your article has been reviewed by three peer reviewers, and the evaluation has been overseen by Nick Turk-Browne as the Reviewing Editor and Michael Frank as the Senior Editor. The following individual involved in review of your submission has agreed to reveal her identity: Dori Derdikman.

Summary:

The authors test the involvement of entorhinal cortex in covert shifts of spatial attention. As with locomotion and gaze, but now without movements, entorhinal neurons fired in a grid-like manner while macaques attentionally tracked a moving target during fixation. These findings suggest a broader cognitive role for entorhinal cortex in mapping and tiling different spaces. The reviewers agreed that this was an interesting and careful study. At the same time, they raised several concerns, including about potential eye movements, links to behavior, and data reporting and visualization. The reviewers discussed the reviews with each other and the Reviewing Editor has drafted this decision to help you prepare a revised submission. We are optimistic about the suitability of this manuscript for *eLife* and hope you will be able to submit the revised version within two months.

Essential revisions:

1) In contrast to earlier work by this group (Killian et al., 2012), the authors attribute the present findings to covert, not overt spatial attention. Yet, they do not report a systematic overview on fixation performance or eye movements during the task. The monkey was trained to keep fixation within a 4.5x4.5 degree window, but could the present findings be explained by (even small) eye movements if they were correlated with the movement of the attention target? If so, fixation accuracy should depend on the distance between the fixation cross and the attention target. A thorough examination of eye movement/fixation performance during the task seems critical to exclude these possible confounds.

2) Can the authors differentiate between high-attention and low-attention trials, either based on behavior (e.g., correct vs. incorrect detection or before vs. after detection) or based on classical attention measures (e.g., pupil size)? Does average activity of grid cells differ between these trials or does it correlate with detection performance across trials (e.g. shorter response times in trials with stronger grid cell activity)? The present task seems well-suited to examine links to behavior, which, while not challenging the key claims of the paper, could yield insights into the involvement and functional significance of the EC in these processes.

3) Visual acuity typically decreases from center to periphery of the visual field as receptive field sizes increase. In addition, most visual areas mainly encode the contralateral field of view. Do the authors see any such relationship between entorhinal receptive fields (e.g. size) and their position in the visual field? Despite differences in perceptual acuity, the attention target seemed to have elicited similar sized entorhinal receptive fields across visual space (unlike in visual cortex). Relatedly, were there hemispheric differences in EC responses?

4) The reported percent of grid cells during covert attention (14%) seems somewhat low compared to rats (e.g., Sargolini et al., 2006; although comparable to Killian et al.). This leaves open the possibility that grid cells during navigation and movement are more prevalent. If true, this might suggest a diminished role of EC in covert attention. The low percentage should be discussed more extensively, along with whether and how this might weaken the conclusions.

5) The authors compare against a place cell, random, and grid cell model. Inspecting some of the grid cells in the methods figure, however, it appears that some of them may simply be multi-peaked place fields (a la Andre Fenton and Loren Frank). It would be useful to pit this model of multi-peaked place fields with no grid structure against the data. For example, could it be that other subsets of EC cells simply fire at multiple locations?

[Editors' note: further revisions were requested prior to acceptance, as described below.]

Thank you for resubmitting your work entitled "Entorhinal cortex receptive fields are modulated by spatial attention, even without movement" for further consideration at *eLife*. Your responses and revisions were comprehensive and compelling, and so we are pleased to accept your article in principle.

However, before that can happen, you need to correct an error. Specifically, in the changes made to address questions about laterality and eccentricity, the letter and manuscript report different statistical results:

Letter: "The first quantifies the difference in firing rate between the left and right side of the visual field. While cells classified as grid cells (5% FDR) in both animals showed, on average, increased firing rates on the ipsilateral side of the visual field relative to the recorded hemisphere, these differences were not consistent across cells (t=-1.4, p=0.18, one sample t-test). Second, we computed an index that compares average firing rate within the central five degrees of visual field around the fixation cross with the rest of the visual field. We again find no difference in firing rate (t=0.06, p=0.95)."

Manuscript: "The first quantifies the difference in firing rate between the right and left side of the visual field. While cells classified as grid cells (5% FDR) in both animals showed, on average, increased firing rates on the contralateral side of the visual field relative to the recorded hemisphere, these differences were not consistent across cells (t=-1.99, p=0.06, one sample t-test). Second, we computed an index that compares average firing rate within the central five degrees of visual field around the fixation cross with the rest of the visual field. We again found no difference in firing rate (t=1.7, p=0.10)."

Moreover, the direction of the laterality effect flips from ipsilateral to contralateral. Finally, if the statistics for the eccentricity effect are correct in the manuscript (trending), then you should likewise report the direction of that effect.

To be clear, both patterns of results lead to the same interpretation and so correcting this will not affect acceptance. However, even if the manuscript is currently correct, the letter will be published alongside, making the inconsistency a matter of public record.

---

## [Author Response]

Essential revisions:1) In contrast to earlier work by this group (Killian et al., 2012), the authors attribute the present findings to covert, not overt spatial attention. Yet, they do not report a systematic overview on fixation performance or eye movements during the task. The monkey was trained to keep fixation within a 4.5x4.5 degree window, but could the present findings be explained by (even small) eye movements if they were correlated with the movement of the attention target? If so, fixation accuracy should depend on the distance between the fixation cross and the attention target. A thorough examination of eye movement/fixation performance during the task seems critical to exclude these possible confounds.

We do not find a correlation of eye-movements and dot position.

This is an excellent point that we address with a new analysis. We have, for every session, correlated the distance of eye position and dot position to the fixation cross and compared this to correlations where we shuffled trials in time. This destroys the correct relationship between eye-movements and dot position. We find that empirical eye-dot correlations are very small (<0.01) and well within the range of what to expect based upon shuffled trial sequences. We therefore have no indication that our results can be explained by overt attention changes. This analysis is included in the revised manuscript.

2) Can the authors differentiate between high-attention and low-attention trials, either based on behavior (e.g., correct vs. incorrect detection or before vs. after detection) or based on classical attention measures (e.g., pupil size)? Does average activity of grid cells differ between these trials or does it correlate with detection performance across trials (e.g. shorter response times in trials with stronger grid cell activity)? The present task seems well-suited to examine links to behavior, which, while not challenging the key claims of the paper, could yield insights into the involvement and functional significance of the EC in these processes.

We agree with the reviewers that the link between grid cell activity and behavior is an important question. We computed average firing rates for cells that were classified as grid cells (based on different thresholds) before correct responses, early or late responses, and fixation errors. We do not find significant differences in these firing rates. However, in the light of the structure of the task this is not too surprising: First, the performance of the monkeys is rather high (85% correct on average) yielding very few error trials. Second, the spatial location of the dot is irrelevant for detecting the color change, which can very likely be carried out by early visual cortex neurons. Therefore, we refrain from strong claims based on the present data, but in the revised discussion, we suggest a key modification of the task design that could address this question in future studies.

3) Visual acuity typically decreases from center to periphery of the visual field as receptive field sizes increase. In addition, most visual areas mainly encode the contralateral field of view. Do the authors see any such relationship between entorhinal receptive fields (e.g. size) and their position in the visual field? Despite differences in perceptual acuity, the attention target seemed to have elicited similar sized entorhinal receptive fields across visual space (unlike in visual cortex). Relatedly, were there hemispheric differences in EC responses?

We find no significant eccentricity or hemispheric differences

Systematic biases in receptive field size and position caused by hemisphere or eccentricity specific processing should show up as systematic biases in firing rate across the visual field. We therefore computed two asymmetry indices. The first quantifies the difference in firing rate between the left and right side of the visual field. While cells classified as grid cells (5% FDR) in both animals showed, on average, increased firing rates on the ipsilateral side of the visual field relative to the recorded hemisphere, these differences were not consistent across cells (t=-1.4, p=0.18, one sample t-test). Second, we computed an index that compares average firing rate within the central five degrees of visual field around the fixation cross with the rest of the visual field. We again find no difference in firing rate (t=0.06, p=0.95). These numbers, however, have to be taken with a grain of salt. In each monkey we recorded one hemisphere only, making an intra-monkey comparison of left/right visual field responses difficult. Furthermore, the experiment was not designed to optimize investigation of systematic changes with eccentricity. For such an experiment, monkeys trained to covertly track targets at high eccentricity and as well as more recording time for each neuron with attention directed to eccentric locations would be desirable. These two constraints interact and make respective experiments highly challenging.

In the revised manuscript we included this analysis and observation and make a suggestion for future studies.

4) The reported percent of grid cells during covert attention (14%) seems somewhat low compared to rats (e.g., Sargolini et al., 2006; although comparable to Killian et al.). This leaves open the possibility that grid cells during navigation and movement are more prevalent. If true, this might suggest a diminished role of EC in covert attention. The low percentage should be discussed more extensively, along with whether and how this might weaken the conclusions.

We see two possible explanations for the lower ratio of grid cells compared to rats. First, our task design is limited to detecting a subset of grid cells in which the firing fields match the size of the screen and the spacing of the dot’s trajectory. Second, our task design does not attach significance to the spatial position of the dot. It is possible to solve the task without knowing where the color change occurs. Thus, it seems plausible that we observe the entorhinal cortex in an attenuated state and that we would observe stronger grid cell activity if the exact dot position was relevant. It is also difficult to directly compare percentages across species, given the significant differences in recording techniques. For example, we did not preferentially record from layer 2 neurons. We believe that these possibilities do not weaken our conclusion; however, we discuss these aspects in the revised manuscript and hope that future work will unravel these distinct possibilities.

5) The authors compare against a place cell, random, and grid cell model. Inspecting some of the grid cells in the methods figure, however, it appears that some of them may simply be multi-peaked place fields (a la Andre Fenton and Loren Frank). It would be useful to pit this model of multi-peaked place fields with no grid structure against the data. For example, could it be that other subsets of EC cells simply fire at multiple locations?

We thank the reviewers for pointing us to the work by Andre Fenton and Loren Frank. However, the question raised is a bit tricky. The grid cells we report in the present manuscript are differentiated from place cells by multiple peaks of activity and a regular arrangement of these peaks. Thus, the putative multi-peaked place cells would form a superset of the grid cells.

The difference between true grid cells and multi-peaked place cells without a grid structure would be captured by the grid score as well as the modulation index. Obviously, for the latter cell type the grid score would be lower as the peaks are not arranged in a grid-like pattern. Yet, the modulation index is independent of the layout of the firing field and would therefore be of comparable size to grid cells. Thus, in Figure 2 such cells would be found in the center of the display. Essentially, this means that units in the border region between place cells and grid cells would comprise the multi-peaked but not grid cell class. The precise placement, however, depends highly on the degree of irregularity. Minute variances would make them very similar to grid cells, high variability would position this class lower (and to the left) in the place cell region.

Detecting smaller irregularities that can be judged only when viewing a vast extent of the grid structure is possible in the rodent hippocampus. There, many complete cycles of the grid can be observed, facilitating detection of deviations from a regular structure. With the smaller ratio of field of view and cycle length of the grid-like structure observed in primates this is more difficult. Thoroughly differentiating multi-peaked place fields from grid-cells require additional parameters for spatial variations of peaks that can easily lead to overfitting with our limited field of view. Therefore, we think a quantitative evaluation of this aspect is of limited use.

Given that *eLife* publishes (in case the manuscript is accepted) this reply jointly with the main manuscript, this issue is documented. However, we feel uneasy to add this material to the main manuscript. Instead, we fully document the form of the firing fields in many examples in different regions in Figure 2. We hope that this allows for an unbiased appraisal of the spatial structure of those neurons.

[Editors' note: further revisions were requested prior to acceptance, as described below.]

However, before that can happen, you need to correct an error. Specifically, in the changes made to address questions about laterality and eccentricity, the letter and manuscript report different statistical results:Letter: "The first quantifies the difference in firing rate between the left and right side of the visual field. While cells classified as grid cells (5% FDR) in both animals showed, on average, increased firing rates on the ipsilateral side of the visual field relative to the recorded hemisphere, these differences were not consistent across cells (t=-1.4, p=0.18, one sample t-test). Second, we computed an index that compares average firing rate within the central five degrees of visual field around the fixation cross with the rest of the visual field. We again find no difference in firing rate (t=0.06, p=0.95)."Manuscript: "The first quantifies the difference in firing rate between the right and left side of the visual field. While cells classified as grid cells (5% FDR) in both animals showed, on average, increased firing rates on the contralateral side of the visual field relative to the recorded hemisphere, these differences were not consistent across cells (t=-1.99, p=0.06, one sample t-test). Second, we computed an index that compares average firing rate within the central five degrees of visual field around the fixation cross with the rest of the visual field. We again found no difference in firing rate (t=1.7, p=0.10)."Moreover, the direction of the laterality effect flips from ipsilateral to contralateral. Finally, if the statistics for the eccentricity effect are correct in the manuscript (trending), then you should likewise report the direction of that effect.To be clear, both patterns of results lead to the same interpretation and so correcting this will not affect acceptance. However, even if the manuscript is currently correct, the letter will be published alongside, making the inconsistency a matter of public record.

We are sorry for submitting a revision where letter and manuscript were inconsistent. The submitted manuscript was correct, while the letter contained a statement about an erroneous analysis. In particular, during the analysis we made an error when coding response hemispheres for individual cells, assigning cells to essentially random hemispheres. We identified this error during our internal review, but then apparently only updated the manuscript. We’ve now updated the manuscript to also report the magnitude and direction of both asymmetry indices. We are truly sorry for causing trouble and extra work.